# KLIP: Keyword-Guided Language-Image Pre-training for Data-Efficient Domain-Specific Image Captioning

## Abstract

Image captioning aims to generate natural language descriptions for a given image. While recent vision-language models have shown promising progress on this task, it is still challenging to fine-tune such models for particular domains with limited image-caption training data. To enable domain-specific few-shot image captioning, we propose a Keyword-Guided Language-Image Pretraining (KLIP) scheme, which learns entity-oriented keywords for aligning visual and textual modalities in each data domain for pre-training and fine-tuning. While our pre-training objectives enables the above alignment for vision-language models, the identified keywords further serve as prompts for regularizing the model during the fine-tuning stage. As a result, potential overfitting problems can be mitigated. Extensive experiments on benchmark datasets show that our KLIP performs favorably against state-of-the-art VLMs with various parameter-efficient fine-tuning techniques for domain-specific yet data-efficient image captioning.

## 1 Introduction

Image captioning has achieved remarkable progress across computer vision and natural language understanding, facilitating a wide range of applications such as multimedia retrieval (Cao et al., 2022; Wang et al., 2020) and enhancing accessibility for the visually impaired (Gurari et al., 2020; Ahsan et al., 2021). Despite these advances, existing benchmarks (*e.g.,* COCO Captions (Chen et al., 2015) and Flickr30K (Young et al., 2014)) largely focus on general domain captioning, overlooking the distinct requirements of real-world domain-specific scenarios such as sports commentary (Yu et al., 2018; Hammoudeh et al., 2022) and medical assistance (Pavlopoulos et al., 2019; Liu et al., 2021). While sharing the *same* visual concepts with COCO Captions or Flickr30K, the above scenarios often require more detailed and specialized descriptions unique to the individual domains. As a result, model fine-tuning for the downstream task becomes necessary.

Recently, vision-language foundation models (VLMs) (Li et al., 2020; Zhang et al., 2021a; Chen et al., 2020; Kim et al., 2021; Li et al., 2021; 2022; 2023) have achieved great success on various multimodal downstream tasks. Leveraging web-scaled pretraining data (*e.g.,* Conceptual Captions (Changpinyo et al., 2021) and LAION-5B (Schuhmann et al., 2022)), VLMs effectively align information across both image and text modalities, so that the trained model could be seamlessly fine-tuned to downstream tasks required jointly vision and language understanding (*e.g.,* image captioning, image-text retrieval, and visual question answering). For instance, BLIP (Li et al., 2022) enables vision and language alignment by enforcing the derived representations from paired image and text to be similar, and predicting whether a pair of image and text is matched. Despite their promising performance, VLMs often require a *sufficient* amount of training data in the fine-tuning process to be effectively transferred to downstream tasks. However, in a specific domain, collecting such a large dataset is not always feasible. And, fine-tuning the entire VLMs with limited data can lead to overfitting. Therefore, finding an effective way to fine-tune VLMs with *few-shot* domain-specific data is a critical challenge.

To address this challenge, current VLMs have explored various strategies to perform few-shot learning. First, in-context learning (ICL) (Tsimpoukelli et al., 2021; Alayrac et al., 2022) allows a pretrained model to perform novel downstream tasks without additional fine-tuning by integrating training examples directly into the input. However, ICL incurs significant computational costs by handling

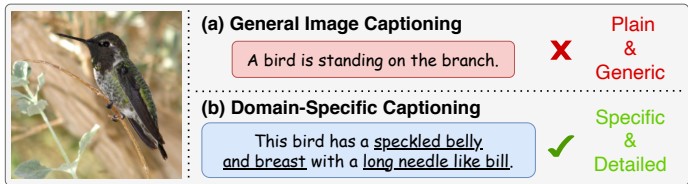

Figure 1: Comparisons between general and domain-specific image captioning.

all training examples for every prediction and often yields sub-optimal performance compared to parameter-efficient fine-tuning (PEFT) methods (Liu et al., 2022). For PEFT, it aims to alleviate overfitting during few-shot fine-tuning by updating only few newly-introduced parameters (e.g., adapters (Sung et al., 2022b; Lu et al., 2023) or prompt-learning based methods (Jia et al., 2022)). Selective methods (Gheini et al., 2021; Zaken et al., 2021) fine-tune a subset of the existing model parameters (*e.g.,* cross-attention layers or bias-terms of the network). Despite its popularity, the exact architecture design of parameter choices typically require users to determine during PEFT.

In this paper, we focus on *Data-efficient Domain-specific Image Captioning*, which involves pretraining on general domain data (*e.g.,* COCO Captions (Chen et al., 2015)), followed by fine-tuning to the targeted specific domain (*e.g.,* CUB (Wah et al., 2011)) using only few-shot samples. It is worth repeating that, as depicted in Fig. 1, domain-specific image captioning requires more detailed and specific descriptions while sharing the same visual concepts with the general/source domain. To address this fine-tuning task, recent approaches (Chen et al., 2017; Zhao et al., 2020) utilize *unpaired* image and caption target-domain data; however, they still need to collect a large amount of unpaired image and caption data. Instead, we focus on using only *few-shot* image-caption pairs in the targeted specific domain during the fine-tuning stage, offering a realistic solution for real-world scenarios.

In this paper, we propose a novel *Keyword-Guided Language-Image Pretraining* (KLIP) scheme that identifies entity-oriented keywords from image-caption data, which describe entity-level visual concepts for aligning visual and textual modalities in each domain of interest. Once such domain-specific keywords are observed, fine-tuning the pretrained model solely requires updating the parameters in the last few textual feedforward layers. This parameter-efficient fine-tuning strategy not only mitigates possible overfitting issues but also improves the transferability to the targeted specific domain. To achieve this, we employ a *KeyWord-Query TransFormer* (KW-QFormer) architecture together with *keyword-aware cross-modal pretraining* tasks during pretraining. The former advances a Transformer-based architecture to extract keyword-aware visual features from input keyword queries through the cross-attention mechanism, while the latter enforces the KW-QFormer to align the visual and textual features conditioned on the guidance of entity-oriented keywords. Through KLIP pretraining, few-shot finetuning for domain-specific image captioning would be successfully enabled with adequate fidelity and adequacy.

The contributions of this paper are highlighted as below:

- We propose a *Keyword-Guided Language-Image Pretraining* (KLIP) scheme, which learns entity-oriented keywords for aligning visual and textual information, so that data-efficient fine-tuning in the downstream domain can be achieved.

- We introduce a novel network module of Keyword-Query Transformer (KW-QFormer) with keyword-aware pre-training tasks, which equip visual-language models with the ability to align visual and textual modalities.

- During few-shot finetuning, we prompt the associated domain-specific keywords which regularize visual-related modules and update a small amount of parameters (*e.g.,* 4.9%) in textual feedforward layers for domain-specific image-caption pairs.

## 2 RELATED WORKS

**Image Captioning**    Recent advancements in image captioning mainly focus on designing different model architectures (*e.g.,* attention mechanisms) (Wang et al., 2019; Huang et al., 2019; Guo et al., 2020; Cornia et al., 2020; Zhang et al., 2021b) and exploring types of representations to describe vision and language modalities (Luo et al., 2021; Kuo & Kira, 2022; Wu et al., 2022; Nguyen et al., 2022). However, the above methods are on COCO Captions (Chen et al., 2015) or Flickr30K (Young

et al., 2014) for object-level captioning in general domain, which cannot generalize well to specific domains (*e.g.,* CUB or medical domain). Recent approaches (Chen et al., 2017; Zhao et al., 2020) consider cross-domain image captioning to adapt the model trained on the general domain (*e.g.,* COCO Captions (Chen et al., 2015)) to the specific domain with unpaired image and text data. Nevertheless, collecting massive unpaired image and text samples is still challenging and potentially impractical.

**Vision-Language Pretraining (VLP)** Recently, leveraging vision-language pretraining (VLP) models learned with web-scale image-text pairs (*e.g.,* Conceptual Captions (Changpinyo et al., 2021) and LAION-5B (Schuhmann et al., 2022)) to improve the performance of vision and language downstream tasks has become a prominent paradigm. With the aim of enabling cross-modal understanding from both visual and linguistic modalities, several types of pretraining tasks are introduced in VLP literature (Li et al., 2020; Zhang et al., 2021a; Chen et al., 2020; Kim et al., 2021; Li et al., 2022; 2023; 2021). For example, UNITER (Chen et al., 2020) and ViLT (Kim et al., 2021) apply masked language/region modeling as the pretext task that predicts the masked tokens from both modalities, and Oscar (Li et al., 2020) and VinVL (Zhang et al., 2021a) additionally utilize object tags to facilitate semantic alignments between paired images and texts. To explicitly empower the text generation tasks, BLIP (Li et al., 2022) and BLIP-2 (Li et al., 2023) additionally adopt language modeling loss to encourage the model to generate textual descriptions given an image. While zero-shot capabilities are observed from VLP models, they tend to produce general captions without sufficient adequacy on the target domain. Recent work VisualGPT (Chen et al., 2022) demonstrates that fine-tuning parameters of foundation models lead to performance improvement. However, fine-tuning such large-scale VLP models using limited paired data often results in severe overfitting issues.

**Parameter-Efficient Fine-Tuning (PEFT)** In recent years, as model size significantly increased, PEFT has emerged as an efficient approach for fine-tuning a large pre-trained model by training only a minimal subset of model parameters. Moreover, when the model's complexity is reduced, the possible overfitting issue during few-shot fine-tuning can be seamlessly mitigated (Ying, 2019). Inspired by the pioneering works (Houlsby et al., 2019; Lester et al., 2021; Li & Liang, 2021; Hu et al., 2021; Zaken et al., 2021) in NLP domain, researchers have also applied these techniques in the vision domain (Sung et al., 2022b; Yang et al., 2022; Zhou et al., 2022b;a; Jia et al., 2022; Bahng et al., 2022; Sung et al., 2022a). For instance, adapters (Sung et al., 2022b; Yang et al., 2022) insert small bottleneck modules into the transformer layers of the pre-trained model, and only fine-tune these adapters along with layer normalization parameters. Similarly, the prompt-based methods (Zhou et al., 2022b;a; Jia et al., 2022) prepend a series of trainable tokens either at the input sequence or within the intermediate layers, with all other parameters keep frozen. Instead of inserting new modules or parameters, selective methods (Gheini et al., 2021; Zaken et al., 2021) fine-tune a subset of the existing model parameters, such as cross-attention layer (Gheini et al., 2021) and bias-term (Zaken et al., 2021). On the other hand, our proposed method can be viewed as prompt-regularized PEFT, which aligns visual and textual information and updates a small set of parameters of interest.

## 3 PROPOSED METHOD

### 3.1 PROBLEM FORMULATION

We first define the setting and notations of our work. Given a large-scale pre-training dataset $\mathcal{D}_p = \{(I_p^i, T_p^i)\}_{i=1}^{|\mathcal{D}_p|}$ with image-caption pairs in general (source) data domains, our goal is to pre-train a captioning model $M_\theta$ on $\mathcal{D}_p$ which could be effectively fine-tuned to a specific target domain $\mathcal{D}_t = \{(I_t^i, T_t^i)\}_{i=1}^{|\mathcal{D}_t|}$ with only a small amount of image-caption data pairs observed (*i.e.,* $|\mathcal{D}_t| \ll |\mathcal{D}_p|$). Note that, following Chen et al. (2017); Zhao et al. (2020), while the target domain shares the same visual concepts with the general domain, it possesses a distinct domain-specific captioning style. To tackle this task, we propose a *Keyword-Gguided Language-Image Pretraining* (KLIP) scheme, which identifies entity-oriented *keywords* for aligning vision and linguistic modalities for pretraining $M_\theta$. Once the pretraining of $M_\theta$ is complete, we leverage the keywords across data domains and perform data-efficient fine-tuning, so that $M_\theta$ can be applied for captioning in the target domain of interest. Thus, how to learn keywords describing domain characteristics for allowing few-shot fine-tuning would be the task to be addressed.

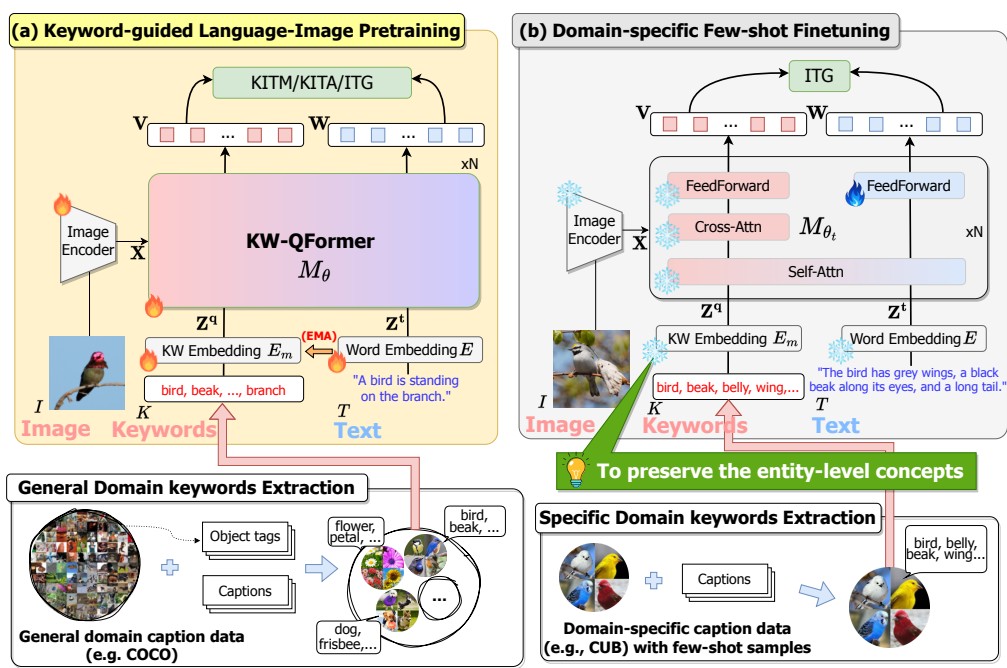

Figure 2: (a) Keyword-Guided Language-Image Pretraining (KLIP) exploits entity-level keywords in a general source domain for pretraining (Sec. 3.2). (b) Domain-specific few-shot finetuning utilizes keywords from the associated domain for finetuning a small amount of parameters of interest (Sec. 3.3). Note that the notation of blue flame indicates that only the last few layers are trainable.

## 3.2 KEYWORD-GUIDED LANGUAGE-IMAGE PRETRAINING (KLIP)

### 3.2.1 ENTITY-ORIENTED KEYWORD DISCOVERY FROM GERNERAL IMAGE-CAPTION DATA

Given a large-scale image-caption dataset for pertaining, the first goal of our KLIP is to discover the entity-oriented keywords, which describe domain characteristic information. As depicted in the bottom of Fig. 2(a), we separate the large-scale pretraining data $\mathcal{D}_p$ into multiple pseudo domains and extract keywords from them via LDA (Hoffman et al., 2010). To be more specific, we take object tags $\{o_p^i\}_{i=0}^n$ from each $I_p$ using VinVL (Zhang et al., 2021a), and we concatenate $T_p$ with $\{o_p^i\}_{i=0}^n$ into textual representation. LDA utilizes a probabilistic Dirichlet-multinomial framework to divide such tag-caption representations into different domains, with each domain additionally outputting keywords with the highest posterior probability. In other words, the output of LDA is represented as $\{(\widetilde{\mathcal{D}}_p^1, K_1), (\widetilde{\mathcal{D}}_p^2, K_2), ..., (\widetilde{\mathcal{D}}_p^N, K_N)\}$, where $\widetilde{\mathcal{D}}_p^i$ is the $i$-th domain, $K_i = \{k_{i,j}\}_{j=1}^M$ is the associated keywords, and $N$ and $M$ are the corresponding domain and keyword numbers, respectively. It is worth noting that the above entity-oriented keywords, derived by integrating both image tag and caption information, are used for aligning the visual and textual modalities in each domain. As discussed later, these keywords additionally serve as prompts for aligning visual and textual feature and for regularizing the model during fine-tuning.

### 3.2.2 KEYWORD-QUERY TRANSFORMER (KW-QFORMER)

As a key network module in KLIP, a unique *Keyword-query Transformer* (KW-QFormer) is proposed to take entity-level semantic information into consideration when relating visual-linguistic data to their associated data domains. As illustrated in the top of Fig. 2(a), our KW-QFormer learns to *query* the keyword-aware visual information from the image encoder through the cross-attention mechanism, while sharing information observed from vision and language modalities by self-attention layers. More precisely, an input image-caption pair $(I, T)$ is encoded to the visual and text features $(\mathbf{X}, \mathbf{Z^t})$ by the image encoder and the word embedding layer $\mathbf{E}$, respectively. For the keywords $K$, they are learned to embed into representations that highly capture the entity-level concepts. Given that the word embedding layer $\mathbf{E}$ has already learned the visual concept through cross-modal learning, we embed $K$ into the keyword representation $\mathbf{Z^q}$ using the keyword embedding layer $\mathbf{E}_m$, which

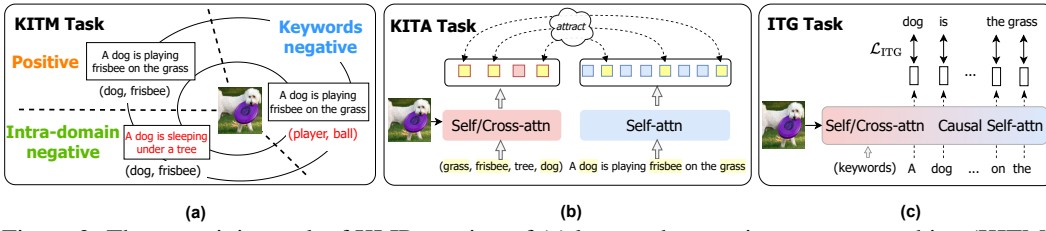

Figure 3: The pretraining task of KLIP consists of (a) keyword-aware image-text matching (KITM), (b) keyword-queried image-text alignment (KITA), and (c) image-ground text generation (ITG) task.

is updated from $\mathbf{E}$ and employs a momentum mechanism to prevent a rapid change of learned information.

As a result, the keyword-aware visual representation $\mathbf{V}$ and the textual feature $\mathbf{W}$ can be derived by our KW-QFormer, denoted as $F_\theta$, where $\theta$ represents the model's parameters. $F_\theta$ allows interaction across multiple-modality features (*i.e.,* visual features $\mathbf{X}$, textual embeddings $\mathbf{Z}^t$, and keyword embeddings $\mathbf{Z}^q$) as follows,

$$\langle \mathbf{V}, \mathbf{W} \rangle = F_\theta(\langle \mathbf{Z^q}, \mathbf{Z^t} \rangle ; \mathbf{X}), \qquad (1)$$

where $\mathbf{V} \in \mathbb{R}^{M \times d}$ and $\mathbf{W} \in \mathbb{R}^{L \times d}$. Here, $M$ and $L$ represents the number of keywords and text tokens, and $d$ denotes the feature dimension, respectively.

### 3.2.3 PRETRAINING KLIP WITH ENTITY-ORIENTED KEYWORDS

**Keyword-aware image-text matching (KITM)**     To enforce the extracted entity-oriented keyword embeddings properly describing semantic concepts across visual and textual modalities, we design three pretraining tasks as follows. The first objective is keyword-aware image-text matching (KITM), aiming to learn the alignment between an image and its corresponding caption based on the keywords of interest. As illustrated in Fig. 3(a), we introduce a 3-way contrastive learning task to encourage the model to distinguish whether the image matches the corresponding caption and keywords.

More precisely, we construct one positive input triplet $\mathbf{P} = (\mathbf{Z^q}, \mathbf{Z^t}, \mathbf{X})$, and two types of negative samples, including *keyword negative* $\mathbf{N^K}$ and *intra-domain negative* $\mathbf{N^I}$. The keyword negative $\mathbf{N^K}$ replaces the keyword embedding $\mathbf{Z^q}$ in $\mathbf{P}$ with mismatched $\mathbf{Z^{q\prime}}$, while the intra-domain negative $\mathbf{N^I}$ is constructed by changing the positive text embedding $\mathbf{Z^t}$ to $\mathbf{Z^{t\prime}}$, which shares the entity-level keywords with $\mathbf{Z^t}$ (*i.e.,* from same domain) but from different image-caption pairs.

As the positive and negative triplets are constructed, we derive keyword-aware visual features $\mathbf{V} = F_\theta(\langle \mathbf{Z^q}, \mathbf{Z^t} \rangle ; \mathbf{X})$ which contain multimodal information to represent the given triplet by allowing cross-modal interaction in the self-attention layers. Hence, with the goal of encouraging the model to predict correctly paired images, captions, and keywords, the keyword-aware image-text matching loss $\mathcal{L}_{\text{KITM}}$ is formulated using cross-entropy loss $\mathbf{CE}$ as follows:

$$\mathcal{L}_{\text{KITM}}(\theta) = \mathbb{E}_{\bar{\mathbf{V}} \sim \{\bar{\mathbf{V}}^{\mathbf{P}}, \bar{\mathbf{V}}^{\mathbf{K}}, \bar{\mathbf{V}}^{\mathbf{I}}\}} \mathbf{CE}(y^{\text{kitm}}, p^{\text{kitm}}(\bar{\mathbf{V}})), \qquad (2)$$

where $\bar{\mathbf{V}}$ comes from averaging $\mathbf{V}$ along with the keyword embedding dimension. $p^{\text{kitm}}$ denotes output probability, and the $y^{\text{kitm}}$ is a 3-dimensional one-hot vector representing the ground-truth label for positive, keyword negative, and intra-domain negative, respectively.

**Keyword-queried image-text alignment (KITA)**     As the second pretraining objective, we propose keyword-queried image-text alignment (KITA) for enhancing fine-grained cross-modal alignment. As depicted in Fig. 3(b), given an input image, KITA aims to align its caption based on the associated keywords. To calculate the image-caption similarity in the *token* level, we measure the similarity $s(\mathbf{V}, \mathbf{W})$ between keyword-aware visual feature $\mathbf{V} = F_\theta(\mathbf{Z^q}; \mathbf{X})$ and the textual feature $\mathbf{W} = F_\theta(\mathbf{Z^t})$ using cosine similarly as:

$$s(\mathbf{V}, \mathbf{W}) = \frac{1}{|\mathcal{P}|} \sum_{p \in \mathcal{P}} \mathbf{V}_{m(p)}^\top \mathbf{W}_p, \qquad (3)$$

where $\mathcal{P}$ is the indices of tokens of interest in text, and $m(p) = \arg\max_r \mathbf{V}_r^\top \mathbf{W}_p$, which indicates the index of most similar keyword given $p$-th text token. We select *nouns* in the caption as the tokens

of interest to align with entity-oriented keywords. As a result, we propose the KITA loss $\mathcal{L}_{\text{KITA}}$ to increase the similarity of the matched image-caption pair in the following,

$$\mathcal{L}_{\text{KITA}}(\theta) = \frac{1}{2}\mathbb{E}_{(\mathbf{V},\mathbf{W})\sim D}\left[\mathbf{CE}(y^{\text{i2t}}(\mathbf{V}), p^{\text{i2t}}(\mathbf{V})) + \mathbf{CE}(y^{\text{t2i}}(\mathbf{W}), p^{\text{t2i}}(\mathbf{W}))\right], \quad (4)$$

where $$p_m^{\text{i2t}}(\mathbf{V}) = \frac{\exp(s(\mathbf{V}, \mathbf{W}^m)/\tau)}{\sum_{i=1}^{M}\exp(s(\mathbf{V}, \mathbf{W}^i)/\tau)}, \quad p_m^{\text{t2i}}(\mathbf{W}) = \frac{\exp(s(\mathbf{V}^m, \mathbf{W})/\tau)}{\sum_{i=1}^{M}\exp(s(\mathbf{V}^i, \mathbf{W})/\tau)}, \quad (5)$$

where $\tau$ is the learnable temperature, and the $\mathbf{W}^i$ and $\mathbf{V}^i$ indicates the $i$-th features in current batch. The ground-truth one-hot vectors are denoted as $y^{\text{i2t}}$ and $y^{\text{t2i}}$, where the positive image-caption pairs are assigned a probability of 1 and the negative pairs a probability of 0.

**Image-grounded text generation (ITG)**   Finally, we apply the objective of the standard image captioning task, as illustrated in Fig. 3(c). Following Li et al. (2022; 2023), we train the model to predict the next token of a caption conditioned on the derived textual features $\mathbf{W} = f_\theta(\langle \mathbf{Z^q}, \mathbf{Z^t} \rangle; \mathbf{X})$ by a linear classifier. Specifically, we define $\mathcal{L}_{\text{ITG}}(\theta)$ to minimize the cross-entropy loss with the ground-truth token id $\hat{y}$:

$$\mathcal{L}_{\text{ITG}}(\theta) = \mathbb{E}_{\mathbf{W}\sim D}\mathbf{CE}(\hat{y}, p^{\text{itg}}(\mathbf{W})). \quad (6)$$

### 3.3 DATA-EFFICIENT FINE-TUNING FOR DOMAIN-SPECIFIC IMAGE CAPTIONING

To finetune the pretrained captioning model $M_\theta$ into $M_{\theta_t}$ using few-shot data in a specific domain, it is necessary to tackle the possible overfitting problem. We uniquely propose prompt-regularized PEFT, which utilizes the keywords of the target domain as domain-specific prompts for regularizing the parameters to be updated.

We now detail our finetuning scheme. As shown in Fig. 2(b), we extract $M$ entity-level keywords $K_s = \{k_{s,j}\}_{j=0}^{M}$ from the specific target domain $\mathcal{D}_s$. As these keywords describe and abstract the unique characteristic of that specific domain, the derived entity-oriented keyword embeddings $Z^q$ serves as a domain-specific prompts, which prompt the KW-QFormer and retain the domain-specific visual features from the image encoder. With these features retained by the domain-specific prompts, we can freeze most of the model parameters regarding keyword-visual interaction, and only fine-tune the last three textual feedforward layers, which account for 4.9% of the total parameters, to learn domain-specific caption styles. Thus, the model parameters $M_{\theta_t}$ only contains these textual feedforward, which can be optimized during fine-tuning using the image-grounding text generation (ITG) loss (Eq. equation 6) to encourage the model to produce captions with adequate fidelity and adequacy on the few-shot specific domain. We define $\mathcal{L}_{\text{ITG}}(\theta_t)$ calculated as follows,

$$\mathcal{L}_{\text{ITG}}(\theta_t) = \mathbb{E}_{\mathbf{W}\sim D}\mathbf{CE}(\hat{y}, p^{\text{itg}}(\mathbf{W})), \text{where} \mathbf{W} = F_{\theta_t}(\langle \mathbf{Z^q}, \mathbf{Z^t} \rangle; \mathbf{X}). \quad (7)$$

It is worth repeating that, through our prompt-regularized strategy, only a small subset of model will be updated. Since the number of model parameters to be updated is significantly reduced, the potential risk of overfitting will be suppressed.

## 4 EXPERIMENTS

### 4.1 DATASETS AND EXPERIMENTAL SETUP

**Datasets**   We consider two pre-training datasets in our experiments. MSCOCO dataset (Chen et al., 2015) comprises about 0.4M image-text pairs. The combined dataset, which is composed of MSCOCO (Chen et al., 2015), Conceptual Captions (Changpinyo et al., 2021) and Flickr30k (Young et al., 2014), is a richer dataset with about 3.5M image-text pairs. As for domain-specific image captioning datasets, we consider CUB-200 (Wah et al., 2011) and Oxford-102 (Nilsback & Zisserman, 2008). The complete CUB-200 (Wah et al., 2011) dataset comprises 11,788 bird images, with each image annotated with 10 captions describing specific and detailed attributes of the birds. Similarly, the full Oxford-102 (Nilsback & Zisserman, 2008) dataset includes 8,189 flower images, each with 10 detailed captions. Further details are described in Sec. A.1.

| Method | Backbone | | Pretrain data size | CUB-200 | | | | Oxford-102 | | | |
|---|---|---|---|---|---|---|---|---|---|---|---|
| | visual | textual | | B@4 | M | C | S | B@4 | M | C | S |
| | | | | | | | **1% fine-tuning data** | | | | |
| VinVL$_{base}$ | X152-C4 | BERT-B | 0.4M | 43.7 | 29.1 | 33.0 | 14.4 | 64.9 | 37.6 | 38.5 | 16.9 |
| VisualGPT | RS101 | GPT-2 | 0.4M | 63.1 | 37.3 | 54.2 | 16.9 | 68.3 | 40.3 | 49.3 | 19.2 |
| BLIP | ViT-B/16 | BERT-B | 3.5M | 64.9 | 37.8 | 63.9 | 17.5 | 70.6 | 41.9 | 54.5 | 19.7 |
| BLIP-2 | ViT-g/14 | OPT-2.7b | 3.5M | 64.6 | 37.7 | 62.1 | 17.1 | 70.4 | 41.6 | 53.9 | 19.5 |
| VinVL$_{large}$ | X152-C4 | BERT-L | 8.9M | 63.2 | 37.5 | 60.0 | 17.8 | 70.1 | 41.9 | 50.3 | 19.5 |
| BLIP | ViT-B/16 | BERT-B | 129M | 66.2 | 38.6 | 65.6 | 18.0 | 73.1 | 42.7 | 61.9 | 20.6 |
| BLIP-2 | ViT-g/14 | OPT-2.7b | 129M | 65.2 | 38.0 | 64.2 | 17.8 | 70.5 | 42.0 | 59.8 | 20.1 |
| InstructBLIP | ViT-g/14 | FlanT5XL | 129M | 66.3 | 38.5 | 65.2 | 17.7 | 73.3 | 42.4 | 62.1 | 20.6 |
| KLIP (Ours) | ViT-B/32 | BERT-B | 0.4M | 64.5 | 37.9 | 62.4 | 17.3 | 70.2 | 42.1 | 55.4 | 19.7 |
| KLIP (Ours) | ViT-B/16 | BERT-B | 3.5M | **66.5** | **39.4** | **70.1** | **18.1** | **73.9** | **43.5** | **65.3** | **21.1** |
| | | | | | | | **5% fine-tuning data** | | | | |
| VinVL$_{base}$ | X152-C4 | BERT-B | 0.4M | 59.0 | 35.7 | 55.1 | 16.8 | 68.5 | 39.7 | 49.1 | 18.7 |
| VisualGPT | RS101 | GPT-2 | 0.4M | 64.1 | 38.0 | 61.8 | 17.2 | 71.1 | 41.3 | 51.3 | 19.6 |
| BLIP | ViT-B/16 | BERT-B | 3.5M | 65.7 | 38.3 | 68.7 | 17.6 | 73.6 | 42.5 | 62.0 | 19.9 |
| BLIP-2 | ViT-g/14 | OPT-2.7b | 3.5M | 63.0 | 38.1 | 68.6 | 17.9 | 73.4 | 42.3 | 60.9 | 19.8 |
| VinVL$_{large}$ | X152-C4 | BERT-L | 8.9M | 64.0 | 38.1 | 65.7 | 17.4 | 73.2 | 41.9 | 60.5 | 19.5 |
| BLIP | ViT-B/16 | BERT-B | 129M | 66.5 | 39.9 | 73.7 | 17.6 | 74.0 | 44.4 | 64.6 | 20.7 |
| BLIP-2 | ViT-g/14 | OPT-2.7b | 129M | 65.9 | 38.9 | 71.1 | 17.9 | 74.1 | 42.9 | 64.8 | 20.0 |
| InstructBLIP | ViT-g/14 | FlanT5XL | 129M | 66.4 | 39.7 | 73.3 | 17.8 | 74.2 | 44.3 | 66.2 | 20.9 |
| KLIP (Ours) | ViT-B/32 | BERT-B | 0.4M | 64.7 | 38.7 | 67.8 | 17.4 | 73.4 | 42.3 | 62.5 | 20.1 |
| KLIP (Ours) | ViT-B/16 | BERT-B | 3.5M | **66.8** | **40.3** | **75.7** | **18.2** | **74.5** | **44.6** | **67.6** | **21.0** |

Table 1: Quantitative results on CUB-200 and Oxford-102 with 1% and 5% data for finetuning. RS101 and X152-C4 denote ResNet101 (Chen et al., 2022) and ResNeXt152-C4 (Zhang et al., 2021a), respectively. B@4, M, C, and S represent BLEU@4, METEOR, CIDEr, and SPICE, respectively.

| Method | Updated Params (%) | CUB-200 | | | | Oxford-102 | | | |
|---|---|---|---|---|---|---|---|---|---|
| | | B@4 | M | C | S | B@4 | M | C | S |
| | | | | | **1% fine-tuning data** | | | | |
| BLIP + VPT | 0.4 | 62.6 | 37.0 | 56.5 | 16.5 | 68.1 | 40.5 | 47.9 | 19.0 |
| BLIP-2 + VPT | 0.4 | 62.8 | 37.2 | 57.6 | 16.7 | 69.2 | 40.9 | 49.1 | 19.3 |
| BLIP + VL-Adapter | 7.6 | 62.9 | 37.3 | 60.6 | 16.8 | 70.2 | 41.5 | 52.9 | 19.6 |
| BLIP-2 + VL-Adapter | 8.3 | 63.0 | 37.5 | 60.8 | 17.0 | 70.7 | 41.8 | 53.2 | 19.8 |
| KLIP (Ours) | 4.9 | **66.5** | **39.4** | **70.1** | **18.1** | **73.9** | **43.5** | **65.3** | **21.1** |
| | | | | | **5% fine-tuning data** | | | | |
| BLIP + VPT | 0.4 | 63.5 | 37.4 | 60.7 | 16.9 | 72.8 | 41.9 | 56.3 | 19.2 |
| BLIP-2 + VPT | 0.4 | 63.6 | 37.4 | 61.5 | 17.1 | 73.0 | 42.2 | 57.6 | 19.4 |
| BLIP + VL-Adapter | 7.6 | 63.9 | 37.7 | 64.3 | 17.0 | 73.2 | 42.3 | 61.2 | 19.7 |
| BLIP-2 + VL-Adapter | 8.3 | 63.5 | 37.9 | 65.9 | 17.4 | 73.7 | 42.6 | 62.3 | 20.2 |
| KLIP (Ours) | 4.9 | **66.8** | **40.3** | **75.7** | **18.2** | **74.5** | **44.6** | **67.6** | **21.0** |

Table 2: Comparisons with Parameter-Efficient Fine-Tuning (PEFT) methods on CUB-200 and Oxford-102 using 1% and 5% data. Note that the updated parameters is the proportion of trainable parameters during fine-tuning compared to those during pre-training.

**Baselines**    We compare our model with several state-of-the-art transformer-based models, including VinVL (Zhang et al., 2021a), VisualGPT (Chen et al., 2022), and BLIP (Li et al., 2022). For all the baselines, we follow their settings and fine-tune the publicly available pretrained checkpoints using few-shot domain-specific data. More implementation details are shown in Sec. A.2.

**Evaluation**    Our model is pretrained on two different pretraining datasets, MSCOCO (Chen et al., 2015) and the combined 3.5M dataset (Chen et al., 2015; Young et al., 2014; Changpinyo et al., 2021), respectively. Note that without further mention, we use the combined 3.5M dataset for pre-training. The pretrained model is then fine-tuned on two domain-specific datasets, CUB-200 Wah et al. (2011) and Oxford-102 Nilsback & Zisserman (2008). For evaluation, we employ the standard evaluation protocol of image captioning tasks. Specifically, we use the BLEU@4 Papineni et al. (2002), METEOR Banerjee & Lavie (2005), CIDEr Vedantam et al. (2015), and SPICE Anderson et al. (2016) as evaluation metrics. For each domain-specific fine-tuning dataset, we report the

| Method | Keyword guidance | Fine-tune method | CUB-200 (1% / 5%) | Oxford-102 (1% / 5%) |
|--------|------------------|------------------|-------------------|----------------------|
| BLIP | - | full | 63.9 / 68.7 | 54.5 / 62.0 |
| BLIP | - | partial | 59.5 / 63.4 | 51.8 / 59.2 |
| KLIP | ✓ | full | 67.9 / 72.8 | 62.3 / 64.3 |
| KLIP | ✓ | partial | **70.1 / 75.7** | **65.3 / 67.6** |

Table 3: Ablation study on fine-tuning strategies in terms of CIDEr. Note that "full" and "partial" refer to fine-tuning the entire model and only the last few textual feedforward layers, respectively.

| Pretraining Tasks | CUB-200 (1% / 5%) | Oxford-102 (1% / 5%) |
|-------------------|-------------------|----------------------|
| ITG | 56.3 / 63.1 | 49.8 / 53.8 |
| ITG + KITM | 59.9 / 64.9 | 52.7 / 58.1 |
| ITG + ITM + ITC | 58.5 / 64.2 | 52.1 / 57.8 |
| KLIP (ITG + KITM + KITA) | **62.4 / 67.8** | **55.4 / 62.5** |

Table 4: Ablation study on the pretraining tasks. Note that ITM and ITC serve as baseline tasks of the proposed KITM and KITA, respectively. We use MSCOCO for pre-training here.

performance on the testing set using 1% and 5% of training data, selecting the checkpoint with the highest CIDEr Vedantam et al. (2015) from the validation set.

## 4.2 QUANTITATIVE RESULTS

**Comparison results on domain-specific image captioning dataset.**    In Table 1, we show that when pre-trained on both the MSCOCO dataset and the combined dataset (*i.e.,* the 0.4M and 3.5M datasets shown in the table), KLIP outperforms all baselines with the same pre-training data when fine-tuning on CUB-200 and Oxford-102. Specifically, when KLIP is pre-trained on the 0.4M MSCOCO, it surpasses VisualGPT's second-best scores by 8.2/6.0 on CUB-200 and 6.1/11.2 on Oxford-102 using 1%/5% fine-tuning data. On the other hand, with the 3.5M combined dataset, KLIP exceeds BLIP's second-best results by 6.2/7.0 on CUB-200 and 10.8/5.6 on Oxford-102 under the same fine-tuning conditions. Notably, even when compared to baselines with 37x more pre-training data (*i.e.,* the 129M shown in the table), KLIP still outperforms them by 1.4 to 4.5 CIDEr on both CUB-200 and Oxford-102 with 1%/5% fine-tuning data.

**Comparison results against PEFT methods.**    Since our KLIP serves as a type of parameter-efficient fine-tuning (PEFT) method, we conducted experiments comparing it with other PEFT methods. In Table 2, we fine-tuned BLIP/BLIP-2 using both Visual Prompt Tuning (VPT) (Jia et al., 2022) and VL-Adapter (Sung et al., 2022b). KLIP outperforms all the baselines when fine-tuning only 4.9% of the parameters on few-shot data. Specifically, on the CUB-200 dataset, with few-shot fine-tuning using 1% and 5% of the data, KLIP exceeds the second-best results from BLIP-2 with VL-Adapter by 9.3 and 9.8 CIDEr scores, respectively. For the Oxford-102 dataset, KLIP surpasses all baselines, besting BLIP-2 with VL-Adapter by 12.1 CIDEr using 1% of the data and by 5.3 CIDEr using 5% of the data.

## 4.3 ABLATION STUDY AND QUALITATIVE ANALYSIS

**Ablation on fine-tuning strategy and pretraining tasks.**    In Table 3, we show the necessity of both keyword guidance and the partial fine-tuning strategy during few-shot fine-tuning. From the table, we observe that the CIDEr scores with keyword guidance are significantly higher than those without, which highlights the importance of utilizing domain-relevant keywords for pretraining and finetuning. Additionally, as the pretrained model is regularized by the specific-domain keywords, we can partially fine-tune a small number of parameters with few-shot data to avoid overfitting. As for Table 3, we illustrates the individual contributions of each pretraining task in KLIP. All of KLIP's pretraining tasks consistently enhance the subsequent fine-tuning results. Notably, our full KLIP model, *i.e.,* ITG, KITM, and KITA, not only achieves the best performance but also surpasses the results from the baseline pretraining tasks, *i.e.,* ITG, ITM and ITC, on both datasets. This underscores the effectiveness of our pretraining strategy.

**Direct vs. our finetuning schemes.**    For qualitative comparisons, we first compare the output captions of BLIP Li et al. (2022) and our KLIP, both finetuned on only 1% of training data of CUB Wah et al. (2011) and Oxford Nilsback & Zisserman (2008), respectively. As illustrated in Figure 4, our output captions are more detailed and accurate compared to BLIP.

**Keyword-attended captions and image regions.**    To visualize the extracted entity-oriented keywords in the target domain, we provide examples in Figure 5. In this figure, each of our generated

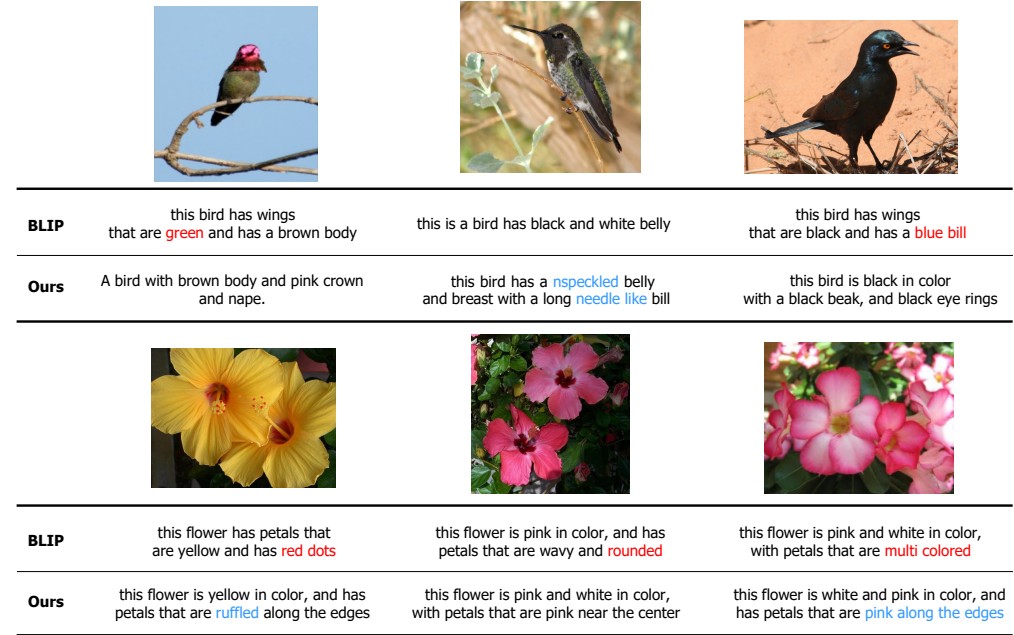

Figure 4: Example output captions by BLIP and our KLIP. Note that words in red denote incorrect descriptions, while those in blue describe images aligned with the domain-specific caption style in CUB and Oxford. Both BLIP and KLIP are finetuned using 1% of training data.

| Input image | Predicted caption | Attention map | | Input image | Predicted caption | Attention map |
|---|---|---|---|---|---|---|

a small bird with white breast and blue wings with long blue and black tail.

(bird, belly, ..., tail, ...)

**(a)**

this flower is pink in color, and has petals that are ruffled and wavy.

(flower, petal, stamen, ...)

**(b)**

Figure 5: Visualization examples of input image, entity-oriented keywords, predicted caption, and the attended image regions: (a) CUB and (b) Oxford. Note that the attended entity-oriented keywords and tokens are marked in pink and blue, respectively.

output token can be properly attended by the desirable keywords. In addition, we show that the attended regions of the input images correspond to the aforementioned keywords as well. Take CUB-200 in Figure 5(a) as an example, the tail token (marked in blue) correctly attends to the entity-oriented keyword of tail (marked in pink), which also identifies the corresponding image region (i.e., tail region in the image).

More ablation studies (*e.g.,* number of domains and keywords) and visualization examples can be found in the supplementary (i.e., Sections A.3 and A.4).

## 5 CONCLUSION

In this paper, we proposed Keyword-Guided Language-Image Pretraining (KLIP) which allows visual-language models for domain-specific and data-efficient fine-tuning. By exploiting entity-oriented keywords from image tags and captions, we presented unique pre-training objectives for aligning visual and textual information. More importantly, such keywords further served as prompts to regularize the pre-trained model, which fix and visual embedding modules and update only the last few layers in the textual embedding layers for few-shot fine-tuning. Therefore, potential overfitting problems can be alleviated. We conducted extensive quantitative experiments, verifying that our KLIP performed favorably against state-of-the-art vision-language pretraining methods, while providing sufficient generalization when adapting the fine-tuned model for domain-specific image captioning.

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

## A APPENDIX

### A.1 DATASETS

The first one is the widely used MSCOCO Chen et al. (2015). The MSCOCO dataset is composed of 117,283 images in total, and each image is annotated with 5 manually written captions. For fair comparison with the baseline methods trained on it, we adopt the karpathy training split Karpathy & Fei-Fei (2015), which includes 82,783 images. To demonstrate the scalability of our method, we experiment with a combined dataset that includes MSCOCO Chen et al. (2015), Conceptual Captions Changpinyo et al. (2021), and Flickr30k Young et al. (2014), providing us with a rich resource of about 3.5M image-text pairs. For the finetuning process, we turn to two domain-specific image captioning datasets, namely CUB-200 Wah et al. (2011) and Oxford-102 Nilsback & Zisserman (2008). The CUB-200 dataset contains 11788 bird images and each image is annotated with 10 detailed captions describing specific and detailed attributes of the birds. To create training, testing, and validation sets for this dataset, we use 7,375, 2,933, and 1,480 images, respectively. Similarly, the Oxford-102 Nilsback & Zisserman (2008) dataset contains 8,189 flower images, each annotated with 10 detailed captions. We also divide this dataset into training, testing, and validation sets, with 6,369, 1,155, and 665 images, respectively.

### A.2 IMPLEMENTATION DETAILS

For entity-oriented keyword discovery, we split MSCOCO Chen et al. (2015) and the combined 3.5M dataset Young et al. (2014); Chen et al. (2015); Changpinyo et al. (2021) into 100 and 500 pseudo domains, respectively. The object tags for keywords extraction are from Zhang et al. (2021a). On both datasets, we extract 32 keywords per domain by default. Our KW-QFormer and the word embeddings $E$ are initialized from BERT$_{base}$ Devlin et al. (2019), and the image encoder is initialized from ViT-B/16 pretrained on ImageNet Touvron et al. (2021). Utilizing 8 NVIDIA V100 GPUs and a total batch size of 512, we pretrain our model on MSCOCO for 20 epochs and on the combined 3.5M dataset for 10 epochs. To ensure a sufficient amount of intra-domain negatives for both the KITM and KITA tasks, we sample our training batch from 8 pseudo domains. Each pseudo domain contributes 64 samples to the batch. In the KITA task, to ensure that text tokens can be appropriately matched with keywords, we further enrich our keywords by including object tags from the input image. After pretraining, we fine-tune the model for 40/60 epochs for 5%/1% training data on CUB-200 Wah et al. (2011) and Oxford Nilsback & Zisserman (2008) using a batch size of 64. In both the pretraining and fine-tuning phases, we utilize the AdamW optimizer Loshchilov & Hutter (2017) with a weight decay of 0.05. During pretraining, we warm up the learning rate to 1e-4 over the first 1000 steps for the MSCOCO dataset and the initial 2000 steps for the combined 3.5M dataset, before subsequently decaying it linearly to 5e-6. In the fine-tuning phase, we maintain a constant learning rate of $1e$-5.

### A.3 MORE ABLATION RESULTS

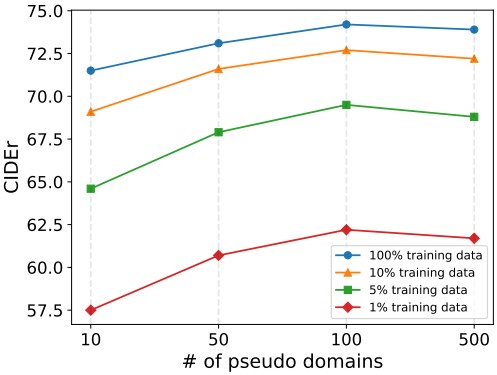
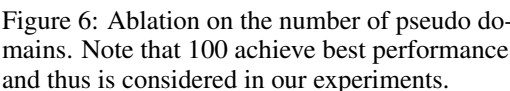
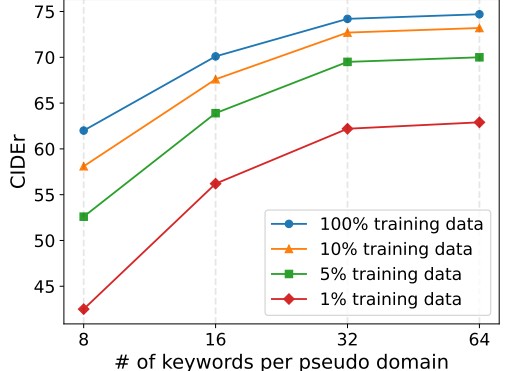

Figure 6: Ablation on the number of pseudo domains. Note that 100 achieve best performance and thus is considered in our experiments.

Figure 7: Ablation on the number of extracted keywords per domain. Note that 32 is sufficient and thus is considered in our experiments.

**Ablation on the number of pseudo domains.** Figure 6 illustrates the influence of the number of pseudo domains, clustered from the source domain dataset, on KLIP's performance. We pretrain KLIP on the MSCOCO dataset Chen et al. (2015) and subsequently fine-tune it on the CUB-200 dataset Wah et al. (2011). The results indicate that the CIDEr score consistently improves as the number of pseudo domains expands from 10 to 100, despite seeing a slight decline at 500. Consequently, we select 100 as the optimal number of pseudo domains for all experiments pretraining on the MSCOCO dataset Chen et al. (2015).

**Ablation on the number of keywords per domain.** Figure 7 demonstrates the impact of the number of keywords on KLIP's performance. For both the CUB-200 and Oxford-102 datasets, the CIDEr score consistently improves as the number of keywords increases from 8 to 64. However, we observe that the performance gains begin to saturate as the keyword count increases. Consequently, to balance computational complexity and performance, we choose 32 for our final parameter setting.

### A.4  MORE QUALITATIVE RESULTS

In Figure 8 and Figure 9, we provide more examples of generated output tokens and their corresponding most-attended keywords, as well as the image regions attended by these aforementioned keywords on CUB-200 Wah et al. (2011) and Oxford-102 Nilsback & Zisserman (2008), respectively. In both figures, the last two images further demonstrate that different keywords attend to different image regions to generate caption tokens. Taking Figure 8(d) as an example, the crown and body tokens (marked in blue) correctly attend to the entity-oriented keywords of "crown" and "body" (marked in pink), and both keywords attend to the corresponding regions (e.g., crown region and body region) in the image, respectively.

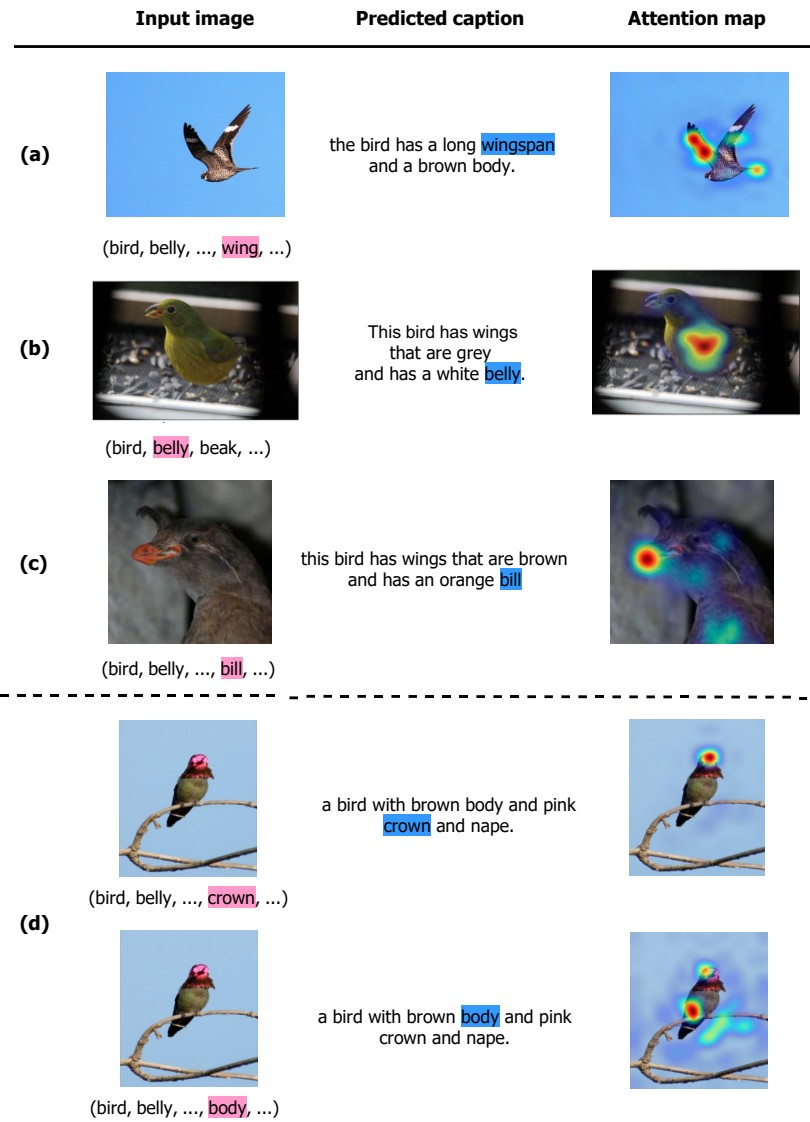

Figure 8: Examples of domain-specific image captioning with input image, entity-oriented keywords, predicted caption, and the attention image region on CUB-200 Wah et al. (2011) dataset. Note that the attended entity-oriented keywords and tokens are marked in pink and blue, respectively.

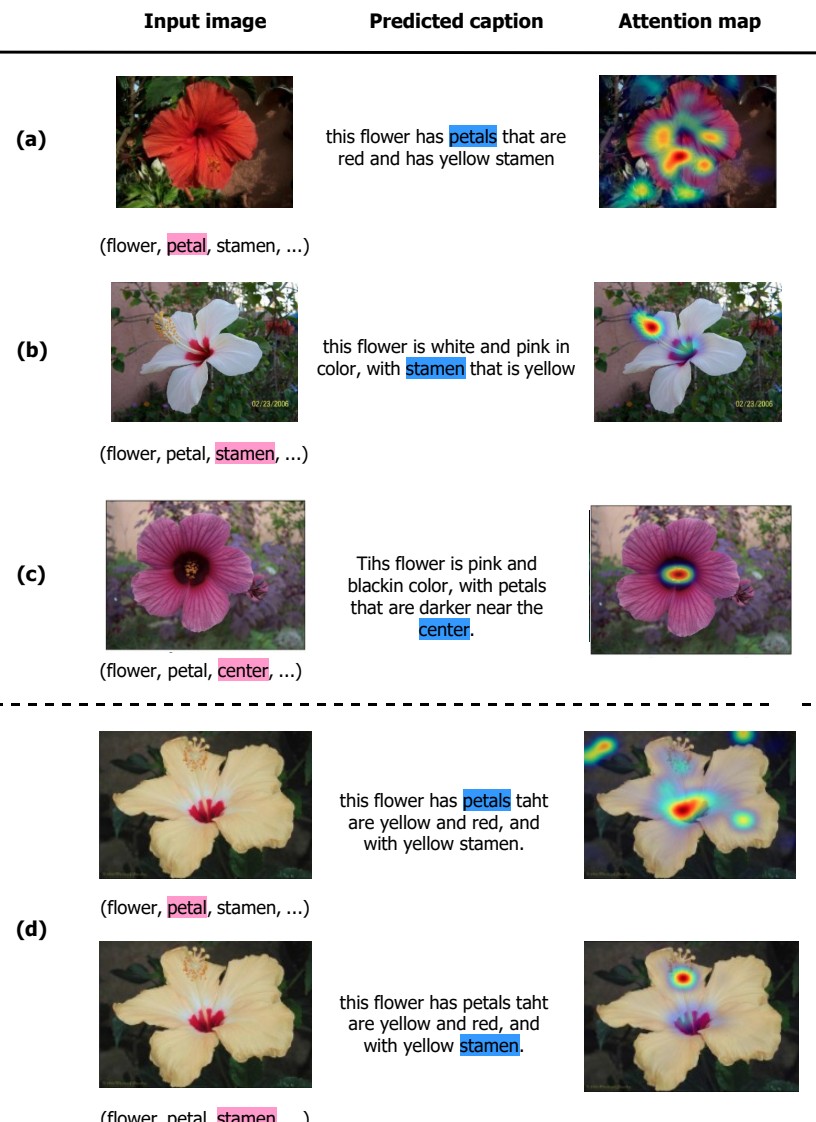

Figure 9: Examples of domain-specific image captioning with input image, entity-oriented keywords, predicted caption, and the attention image region on Oxford-102 Nilsback & Zisserman (2008) dataset. Note that the attended entity-oriented keywords and tokens are marked in pink and blue, respectively.

