# OpenReview forum: "KLIP: Keyword-Guided Language-Image Pretraining for Data-Efficient Domain-Specific Image Captioning"
_ICLR.cc/2024/Conference — ICLR 2024 Conference Withdrawn Submission_

### Official Review · Reviewer_eL8y · 2023-10-25

**Soundness:** 2 fair
**Presentation:** 2 fair
**Contribution:** 2 fair
**Rating:** 5
**Confidence:** 4

**Summary:**

This paper introduces KLIP, a novel vision-language pre-training approach that enhances the model's ability for few-shot adaptation in image captioning on domain-specific datasets.
Specifically, KLIP incorporates keyword information alongside sentences, guided by keyword-aware image-text matching and alignment losses, in addition to the standard image generation loss.
As a result, KLIP surpasses previous methods like BLIP-2 in fine-grained captioning benchmarks such as CUB-200 and Oxford-102.

**Strengths:**

- Enhancing vision-language models for fine-grained tasks represents an important research problem.
- The concept of utilizing keywords to grasp fine-grained information is intriguing yet underexplored.

**Weaknesses:**

**Mixed concept of pre-training and fine-tuning**

The proposed framework blends two distinct concepts: vision-language pre-training (VLP) and parameter-efficient fine-tuning (PEFT).
This obscures the paper's main message and confuses the primary source of improvements.

Here are some subsequent questions:
- What is the effect of KLIP without PEFT? Does KLIP solely improve PEFT, or does it also enhance other representation learning abilities?
- In that regard, what occurs with zero-shot image captioning? Comparing the zero-shot capabilities of BLIP-2/etc. and KLIP would provide better insight into the VLP component, separate from the PEFT aspect.
- The paper should compare the proposed PEFT approach with KLIP + VPT/VL-Adapter instead of BLIP-2 + VPT/VL-Adapter, in order to isolate the effect of the PEFT technique from VLP.
- Other tasks beside image captioning? The proposed VLP is quite versatile, so it could be tested on other tasks such as VQA or image-text retrieval, as shown in Table 1 of BLIP-2.


---
**How do keywords assist in domain-specific captioning?**

The proposed framework regularizes VLP with additional keyword information, which would benefit VLP in general.
However, the claimed motivation, that KLIP is specifically more beneficial for domain-specific image captioning, is not convincing.

Here are more detailed questions:
- Does KLIP also improve captioning on general image domains such as COCO? I believe so; in that case, the proposed method is not specifically designed for domain-specific datasets.
- Since the keywords for pre-training are extracted from the general image domain, they would include general words like "bird" but not many fine-grained terms like "belly." What are the statistics of the keywords for pre-training and fine-tuning?
- If the keywords from pre-training/fine-tuning are similar, the zero-shot ability should be quite good, eliminating the need for fine-tuning. If the keywords are different, why does keyword-guided pre-training benefit fine-tuning?
- Can KLIP generalize to substantial distribution shifts, involving a significant change in keywords? Given its focus on domain-specific adaptation, demonstrating better robustness compared to BLIP-2/etc. would be beneficial.


---
**Additional related work**

This paper [1] explores domain-specific adaptation of a VLP model and considers keyword guidance for this purpose.
However, it employs a CLIP-style training approach, rendering it unsuitable for text generation tasks.
Nevertheless, I believe this paper shares a similar high-level spirit with KLIP.

[1] S-CLIP: Semi-supervised Vision-Language Pre-training using Few Specialist Captions. arXiv'23.

**Questions:**

To summarize the questions above:
1. General representation learning ability of BLIP-2/etc. and KLIP.
2. Zero-shot image captioning abiliaty of BLIP-2/etc. and KLIP.
3. Comparison with KLIP + VPT/VL-Adapter.
4. Beyond image captioning, such as VQA or retrieval.
5. General image datasets such as COCO.
6. Statistics of keywords for pre-training and fine-tuning.
7. Further discussion above depend on the keyword statistics.
8. Robustness on distribution shifts.

---

### Official Review · Reviewer_uxeH · 2023-11-01

**Soundness:** 2 fair
**Presentation:** 2 fair
**Contribution:** 3 good
**Rating:** 5
**Confidence:** 3

**Summary:**

The article proposes Keyword-guided Language-Image Pertaining (KLIP) a pretraining objective that takes into account domain keywords for efficient transfer of a captioning model to specific domains, in a few-shot setting. During training, keywords are extracted by performing unsupervised latent domain discovery (Hoffman et al. 2010) and then collecting them via object tagging (Zhang et al. 2021a). The visual representations are conditioned on the keyword embeddings via a transformer module (i.e. KW-QFormer). The KLIP objective is then composed of three tasks:
1. Image-text generation, the standard captioning objective.
2. Keyword-aware image-text matching (KITM), which constructs positive and negative captions/keywords to push the model to learn alignment between image and text/keyword embeddings.
3. Keyword-queried image-text alignment (KITA) which produces visual features conditioned on tokens of interest (i.e. nouns) in the captions and promotes alignment between these visual features and the corresponding captions, averaging the similarity scores across tokens of interest.

During inference, a small amount of layers processing textual inputs is fine-tuned given the keywords extracted from the few-shot pool of data. Experiments show that this approach outperforms some of the existing VLMs in captioning on fine-grained datasets.

**Strengths:**

1. I find interesting the principle of conditioning a captioning model on the specific domain of interest. This makes the model not only more tailored to specific tasks but also allows for human interventions as, in principle, a user may manually specify the keywords of interest together with the few-shot samples. While this is practically not optimal, it still goes toward the direction of customizable captioning systems, which is not deeply investigated yet.
2. In general, the approach achieves good results on both CUB and Oxford Flowers, surpassing competitive baselines such as BLIP-2 and VinVL (Table 1). Table 2 shows also how, even when coupling BLIP-2 with different types of adapters, still the performance falls behind the proposed approach, showing the effectiveness of the proposed fine-tuning strategy.
3. Table 4 shows how each of the objectives proposed in the work contributes to the final performance, motivating their adoption.
4. The qualitative results in Fig. 4 and Fig. 5 illustrate on how keywords can influence the caption generation process, making the generated text semantically richer and more tailored to the domain of interest.

**Weaknesses:**

I have concerns regarding the presentation of the contribution and some of the design choices. Specifically:
1. Section 3 results unclear in some of its parts. For instance, in Section 3.2.3, it is not clear how the logits are computed (i.e. is the loss binary? it is the dot-product/similarity among representations?) and thus how Eq. (2) is implemented (especially from a scoring perspective).  $M$ indicates both the number of pseudo-domains (Section 3.2.1) and the batch-size (Section 3.2.3). Clarifying details and potential ambiguities is crucial to avoid misunderstandings of the reader and allow an easy use/reproducibility of the approach.
2. While the related work section presents various types of works, it does not highlight how the article contributes to each specific subfield: clarifying this would help in placing the work within the existing literature. Moreover, the section on captioning is slightly limited/unbalanced w.r.t. the other two, coarsely describing related approaches. Expanding it would help the reader understand if the proposed work is the first to use the concept of keywords for captioning or not.
3. The approach relies on two main hyperparameters, namely the number of pseudo-domains and the number of keywords per domain. The former is set to 100 following experiments on MS-COCO (where performance decreases with 500 pseudo-domains). However, as the size of the dataset grows, we may have potentially many more pseudo-domains in the dataset (e.g. for LAION), thus setting this hyperparameter a priori is non-trivial.
4. For the second hyperparameter, it might be that the number of domain keywords actually varies across domains: while approaches like VinVL extract various keywords per image, their number might be much larger in some domains w.r.t. others, and there might be redundancy in fine-grained domains. I find it unclear from the text why fixing the number of keywords is the best choice vs having a dynamic one based on e.g. the number of keywords detected per domain.
5. While BLIP-2 and InstructBLIP are strong baselines, Table 1 does not report results for captioning-specific models (e.g. from Section 2) and even VLM-based ones (e.g. CoCa [a]). These models could provide a more precise picture of the performance of the model vs approaches tailored for the task.
6. The paper stresses the parameter-efficient fine-tuning nature of the approach. However, Table 2 does not show whether the proposed strategy (i.e. fine-tuning three layers on the text feedforward network) is best compared with other ones (e.g. VPT, VL-adapter, as applied on BLIP and BLIP-2). Showing these results would strengthen this claim.

References:\
[a] J. Yu et al., "CoCa: Contrastive Captioners are Image-Text Foundation Models", TMLR 2022.

**Questions:**

While I find the use of keywords interesting, I have some concerns regarding the presentation and analyses of the work. For the rebuttal, I would be interested to read:
1. How is the KITM loss/scoring designed?
2. How does the work compare with specific captioning approaches?
2. How does the method advance the literature presented in Section 2?
3. Is there a particular reason for having a fixed vs a dynamic number of domain keywords?
4. How would a practitioner set the number of pseudo-domains in extremely large-scale scenarios?
5. Is the parameter-efficient fine-tuning approach proposed better than other existing ones applied on KLIP?

**Details Of Ethics Concerns:**

None.

---

### Official Review · Reviewer_g8oh · 2023-11-01

**Soundness:** 2 fair
**Presentation:** 3 good
**Contribution:** 3 good
**Rating:** 5
**Confidence:** 4

**Summary:**

This work proposes a method for sample efficient domain specific image captioning by using in-domain “keywords” as prompt inputs to a transformer module used for image captioning. During pre-training they propose to use LDA for dividing the pre-training datasets into different domains, having different sets of keywords which can be used as input to the KW-Qformer. While finetuning, this process is mimicked, with the input keywords being obtained from the particular domain.

**Strengths:**

1. Novel idea: as far as I can tell, this idea of using keyword inputs is novel and provides a way for better generalization of the captioning task to new domains where fine-tuning data is limited. It also has some analogies to instruction tuning in the LLM space for generalizing to new tasks.
2. The (quality of) execution of the idea seems to be good, using of LDA for dissecting pre-training data into domains and using their keywords while pre-training, using/creating tasks like KITM/KITA which make use of these keywords (creating negative keywords, matching keyword aware visual representations with text features) make sense and help the model to produce better captions.
3. Experimentation is decent, has been done on 2 fine-tuning datasets, 2-pre-training datasets, over 1% and 5% data settings. However, it might not be enough (see weakness Q2,3,6)

**Weaknesses:**

1. It is unclear from this work how well does the model generalize to either 1. New keywords in the fine-tuning set or 2. Unseen combinations of keywords in the pre-training set. This might be the most useful use case of this paper which is unexplored. First step might be to quantify overlap of keywords between pre-training and fine-tuning setup, as discussed in Q3 in questions below. It might be good to see some results on specialized datasets (which may have somewhat a different “domain” than general datasets), for eg, the IU- XRAY [1] dataset used by VisualGPT.
2. While the motivation is domain specific captioning with “less” data, there isn’t a lot of discussion about what less data means and how does it vary across domains. Experiments are performed on 1%/5% of existing datasets, but it is unclear if this translates to any real world scenario (see Question 2 below). For eg. it might be easier to obtain bird images and captions (manually or collecting it from the internet) than it is to obtain X-RAY images and corresponding reports/text.
3. Choice of target domain specific datasets for experimentation is unclear. These datasets are quite old and it is very much possible that information might be leaked from these datasets into the pre-training datasets (especially web-curated ones like the 3.5M dataset used in this work). Given that fine-tuning might. Can there be any measures be taken (may be some sort of filtering) to make sure this domain specific data is different from the pre-training dataset?
4. While pre-training data sizes of all compared methods are listed in Table 1, it doesn't take into account the pre-training datasets used for training the visual and textual backbones of each method. Along with this, these pre-training datasets are important because they can have significant overlap with the domain specific data, and this again raises a question if the model is actually generalizing to the domain specific data, or has some overlap of the domain specific data with the pre-training data (both image-text, image only, and text only pre-training data).
5. Comparisons with other baselines: This work should be compared with baselines such as MAPL [2] which also adapt pre-trained vision and language models for having images as input, thus supporting the captioning task. Given that someone with target domain specific data can choose to use this kind of an approach for their captioning task, leads me to believe that this approach should be a compared with KLIP.

[1] Dina Demner-Fushman et. al, Preparing a collection of radiology examinations for distribution and retrieval.

[2] Oscar Mañas et. al, MAPL : Parameter-Efficient Adaptation of Unimodal Pre-Trained Models for Vision-Language Few-Shot Prompting

**Questions:**

Along with these, please also see questions and concerns mentioned in weaknesses section above.
1. In Table 4, is KITM in the 2nd row the same as ITM in the third row? Is ITC the same as KITA? What is the performance of ITG + KITA?
2. It is assumed that for domain specific fine-tuning, we have “less” data. This is vague. How much low data are we talking about? What is the rationale of testing the approach on 1% or 5% of the data. My concern is that it’s not clear for any domain on how much data one can collect. While it’s good to have performance increase using the proposed approach on 1/5% of the data, there should be more experiments on a range of data fractions, for eg. 20/50% of the data as well for completeness of the study. For eg. 50% of data corresponds to 4-5K samples of the domain specific datasets the paper has considered. 4-5K data points might not be very hard to collect for a domain if one is allowed to also use the internet or collect their datasets from existing image-text datasets.
3. What is the overlap of keywords between the keywords of the target domain, and the keyword of the pre-training data? ie the pre-training keyword sets obtained using LDA, can have some overlap with the fine-tuning keyword set, for eg the keyword “beak” can be present in the pre-training keyword set K_i, and the fine-tuning keyword set K_s. among all the pre-training keyword sets (K_1, K_2, …., K_N) which set does K_s overlap the most with and how much is the overlap? (this question is applicable to combinations of both fine-tuning and all pre-training datasets). This would help us know how much the model is able to generalize to new combinations of keywords in the fine-tuning set. This question encompasses not only the image-text pre-training datasets, but also the image-only and text-only datasets used for vision and text model initialization.

---

### Official Review · Reviewer_NtDh · 2023-11-08

**Soundness:** 2 fair
**Presentation:** 3 good
**Contribution:** 2 fair
**Rating:** 3
**Confidence:** 4

**Summary:**

The paper introduces KLIP as an efficient fine-tuning methods for adapting pre-trained vision-language models for task-specific image captioning. The key contribution is identifying the association of keywords from captions to the visual concepts.

**Strengths:**

- The paper is generally well written and the motivation is clear.
- The model does not require extensive pre-training, i.e., only 3.5M pre-training data.

**Weaknesses:**

- The novelty of the proposed method is limited. The proposed model mainly have an additional input head for keywords. The keyword embedding layer is also borrowed directly from sentence embeddings.
- The proposed method, in principle, should work on conventional captioning datasets, i.e., MSCOCO, Flicker, NoCaps. However, this work failed to compare to the baselines in these datasets.
- Most crucially, the proposed method does not compare to baselines in the setting of full dataset fine-tuning. In fact, the fine-tuning datasets are small already, and studying on the 1% or 5% setting makes less sense. This is also demonstrated by the extreme low CIDEr scores reported in Table 1 and Table 2, etc. Comparing of models like this does not provide meaningful message. Besides, improvements are also marginal, if not visible.

**Questions:**

- There are lacks of implementation details for fine-tuning of the baseline models. -
- Missing mean and std for fine-tuning results (repeated experiments on different portions of the 1% and 5% data).

---

### Official Review · Reviewer_Qycg · 2023-11-09

**Soundness:** 4 excellent
**Presentation:** 3 good
**Contribution:** 4 excellent
**Rating:** 6
**Confidence:** 5

**Summary:**

This paper target the problem of domain specific few-shot captioning. Most prior works have focused on general image captioning domains such as on MSCOCO, where plenty of data is available. This work proposes to perform captioning on specific domains such as those in CUB (birds), Oxford-101 (flowers). They propose a three-stage pipeline where they first extract domain-specific keywords using LDA. They then use these keywords to pre-train a transformer to extract keywords-aware visual embeddings. This transformers computes cross-attention between visual and keyword embeddings and also input the captioning embeddings. The pretraining is done with several objects pertaining to generation and contrastive learning. In the final stage this pretrained model is finetuned in a parameter-efficient way on some specific domain while utilizing the keywords from this domain. The authors show consistent improvements with end-to-end learning based approaches as well as those based on large-scale pretraining. The also show ablation experiments to highlight the benefits of their loss and keyword guidance. Finally the authors also show qualitative results to highlight the difference wrt the previous SOTA method and also attended visual regions from the keywords.

Overall the approach seems novel and is motivated well for this particular task with strong empirical results. There are some missing details and concerns for reproduction

**Strengths:**

- Authors have provided a good motivation for the problem and developed approach
- Overall idea seems simple to follow despite having some bells and whistles to make it work
- Coverage of prior works is good
- Experiments are well designed and try to cover all possible areas such as ablation, benefit of keyword guidance, qualitative results, attention maps etc.
- Figure 2 was very useful to understand the method

**Weaknesses:**

- Paper could have be written well. There were several grammatical mistakes. For example,
"Despite its popularity, the exact architecture design of parameter choices typically require users to determine during PEFT." In Sec1
- Algorithm could have been described better. For example, I wasn't sure if there is a cross-attention between visual and keyword features until I read the method section twice (not clear what is used for query, key ,value). The introduction (esp. the paragraph before contribution) should have been written better as it was hard to parse what this method is doing without reading the entire paper
- Despite a strong method, there were several missing details
  - How is the domain (and thus keywords) determined for an input image during pretraining. Based of my understanding LDA was used over the keywords to get some topics/domains. But then how were they connected with an image. It is not mentioned in the paper.
Again how are the keywords for a domain extracted in the second stage. Are they fixed for all input images? What is the effect of changing these keywords during inference for an image?
  - What do you mean by prompt regularized PEFT? Based on 3.3. I get that these keywords are used by the transformer to extract KW image representation. Is that right? Could you provide an intuition on why is this helpful as compared to not using them? What is unique about the image representations that you extract.
- I might be asking this question again but what is meant by entity-specific keyword? Are the keywords same or different for a domain during inference

**Questions:**

Please see weakness section

---

> ### Author Response · Authors · 2023-11-12
> **Reviews for another work received**
>
> Dear Reviewer Qycg,
>
> We believe that the review you submitted is for another work, since the summary/strengths/weaknesses do not properly describe our work. Please kindly update your review, so that we can correspond to your comments accordingly. Thanks!
>
> Authors of #3480

---

> > ### Comment · Reviewer_Qycg · 2023-11-14
> > **Updated the review**
> >
> > My apologies for the confusion. I have updated the reviews.